# Mutations in the acetylation hotspots of Rbl2 are associated with increased risk of breast cancer

Farman Ullah[1], Nadia Khurshid[1], Qaiser Fatimi[1], Peter Loidl[2], Muhammad Saeed[1]*

**1** Department of Biosciences, Cancer Genetics & Epigenetics Lab, COMSATS University Islamabad (CUI), Islamabad, Pakistan, **2** Institute of Molecular Biology, Innsbruck Medical University, Innsbruck, Austria

* muhammad.saeed@comsats.edu.pk

**Data Availability Statement:** All relevant data are within the manuscript and its Supporting Information files.

**Funding:** The author(s) received no specific funding for this work.

## Abstract

Retinoblastoma like protein-2 (Rbl2) is functionally regulated by phosphorylation and acetylation. Previously, we demonstrated that lysine K1083 (K1079 in human Rbl2) is a potential target for acetylation but its functional role remains elusive. We investigated alterations in human Rbl2 gene specifically targeting exons 19–22 harbouring acetylatable residues *i.e.* K1072, K1083 and K1115 through single stranded conformation polymorphism (SSCP) in breast cancer patients. The K1083 was found altered into arginine (R) in 51% of the cases but K1072 and K1115 remained conserved. The 'K1083R' mutation impairs the acetylation potential of this motif that may result in functional inactivation of Rbl2. These patients also showed poor survival outcome that highlights prognostic relevance of this residue. NIH3T3 cells expressing glutamine (K1083Q) mutated Rbl2 could not be arrested in $G_1$ by serum starvation, whereas cells expressing Rbl2 with K1083R showed prolonged $G_1$ arrest in fluorescence activated cell sorting (FACS) analysis. This suggests that K1083 acetylation is important for $G_1$/S transition. Further, we performed molecular dynamic simulations (MDS) to analyse kinetics of residue K1083 with Cyc-D1/CDK4. Mutations at K1083 impaired this binding exposing neighbouring residues S1080, P1081, S1082 and R1084, hence enhancing the possibility of accelerated phosphorylation. S1080 has previously been reported as a promising candidate of cell cycle dependent phosphorylation in Rbl2. This highlights significance of mutations in the pocket domain of Rbl2 gene in breast cancer, and also strengthen the supposition that K1083 acetylation is pre-requisite for its phosphorylation.

## Introduction

Cell cycle checkpoints are walk through gates in eukaryotes to regulate smooth entry of cells into next phase with high fidelity and accuracy [1]. These surveillance mechanisms enable cells to cease cycling if necessary requirements are not fulfilled [2]. Members of retinoblastoma family (pRb, Rbl1/p107 and Rbl2/p130) are nuclear phosphoproteins. These proteins are known to regulate cell cycle progression through E2F mediated transcription on target promoters, hence play a critical role in neoplastic transformation. Collectively, these proteins are

**Competing interests:** The authors have declared that no competing interests exist.

known as 'pocket proteins'. The term 'pocket' alludes to the presence of a functional pocket domain, which is required for their binding with E2F basal transcription factors and viral onco-proteins [3].

Among pocket proteins, retinoblastoma like protein-2 (Rbl2) has unique attributes to arrest cell cycle. Its expression has been found elevated in quiescent ($G_0$) and differentiated cells, while the founding member 'pRb' maintained its steady levels [4] in these cells. Moreover, human glioblastoma (T98G) [5] and cervical cancer cells (C33A) can only be inhibited by Rbl2 expression [6]. Similarly, nasopharyngeal cancerous cells (HONE-1) showed drastic reduction in Rbl2 expression, while other members kept consistent elevated expression [4].

Rbl2 gene comprises of 22 exons that are mapped to chromosome 16q12.2 flanking over 50 kb of human genomic DNA. This gene encodes a large protein of 1139 amino acids with a molecular weight of 130,000 daltons [7]. The C-terminal domain of the protein is encoded by exons 19–22 [8, 9]. A bipartite nuclear localization signal (NLS) consisting of two clusters of basic residues ($^{1081}$PSKRLR$^{1086}$ and $^{1098}$PTKKRGI$^{1104}$) separated by10 amino acids is also present in the C-terminus of the protein [10].

Activities of Rbl2 proteins are regulated by phosphorylation and acetylation [11, 12]. Rbl2 proteins are abundantly present in hypo-phosphorylated state during $G_1$/S transition of cell cycle and remain bounded to nuclear transcription factors ($E_2F_4$ & $E_2F_5$) via its pocket domain. Upon phosphorylation Rbl2/E2F complexes are dissociated and E2F proteins become available to perform transcription [3, 8]. Acetylation of mouse Rbl2 at K1079, K1068 and K1111 has been reported recently in cell stage specific manner [11]. K1079 in mouse Rbl2 (corresponding to K1083 in human) is present within highly conserved-PSKRLRE- motif in C-terminal domain and had been shown as a prime acetylation site, whereas K1068 and K1111 (human 1072 and 1115 respectively) were found to have lesser acetylation tendency. Previously, phosphorylation of S1080 preceding NLS motif has also been reported [13], which suggests a possible crosstalk mechanism among these posttranslational modifications (PTM). The acetylation at K1083 is also captive because of the fact that previously, pRb acetylation was also reported primarily in the C-terminal domain [14]. A little is known about the exact role Rbl2 acetylation during cell cycle progression, though it has been shown to positively influence phosphorylation [12]. Through this, it has been postulated that these signalling activities of Rbl2 acetylation and phosphorylation might be the functional link in regulating cell cycle events. Given the fact that C-terminus of Rbl2 is involved in interaction with other vital cellular factors like Cyc-D1/CDK4 complex [13], mutations in C-terminus especially in acetylation hotspots and NLS might have serious consequences in carcinogenesis.

Current study was designed to screen genetic alterations within pocket domain and C-terminus (Exon 19–22) harbouring acetylation hotspots and NLS of Rbl2 proteins in breast cancer patients. Functional consequences of these mutations on cell cycle progression were analyzed through flow cytometric analysis of murine cells stably expressing mutated Rbl2 proteins. The impact of these mutations on protein structure and function was further analyzed through *in silico* analysis of binding partners. A mutation at lysine (K) 1083 (corresponds to mouse K1079) was found to be pivotal in association with Cyc-D1/CDK4 complexes. This mutation might have prognostic value and need to be investigated rigorously.

## Material and methods

### Subject enrolment and sample collection

Fresh tissue (n = 200) along with their adjacent normal control tissues (ANCT) and blood samples (n = 200) from the same patients were collected in RNA-Later solution from Holy Family Hospital Rawalpindi, PIMS Islamabad, Leady Reading Hospital Peshawar and Khyber

Teaching Hospital Peshawar at the time of surgery. Blood samples in ethylenediaminetetraacetic acid (EDTA) tubes from age and sex matched healthy individuals (n = 50) were also collected as control. ANCT controls (~ 2cm away from tumor) were clinically identified by oncologists. Current study was conducted after prior approval from the ethical review committee (CIIT-09-10-14) of the COMSTAS University Islamabad. A written consent was obtained from the patients prior to sampling. Sample were stored at -20˚C until further analysis.

## DNA isolation and PCR amplification

DNA from tissues and blood sample were extracted using standard phenol-chloroform method and stored at -20˚C until further analysis. Exon specific primers (S1 Table in S1 File) were used to amplify each exon (19–22) in a separate polymerase chain reaction (PCR) reaction. PCR was performed in a 25 ul reaction mixture containing ~50–100 ng of genomic DNA, 10 mM of each primer, 25 mM dNTPs with 1 unit of Taq DNA polymerase (Solis Biodyne). The PCR conditions were: initial denaturation at 95˚C for 5 minutes entailed by 35 cycles at; 3 min on 95˚ C, annealing for 45 sec at ~ 1˚ C below Tm, extension at 72˚ C for 45 secs; and a final extension at 72˚ C for 10 minutes. The PCR products were visualized on 2% ethidium bromide-stained agarose gel.

## Mutation analysis

Single strand conformation polymorphism (SSCP) analysis was performed on amplified PCR products as described in *Gasser et al.*, 2007 [15] with slight modification. Samples with altered electrophoretic mobility shift along with their controls were re-amplified and were sent for direct sequencing to confirm the nature and type of mutation. The sequencing was performed from commercially available sequencers at Macrogen, Korea. Blast online tool (BioEdit version 7.0.5) was used to align and analyze the sequences.

## Cell culture

NIH3T3 cells (German Collection of Microorganisms and Cell Cultures, No. ACC59) were grown in DMEM/HAMs 15-K medium supplemented with 10% (v/v) fecal calf serum and 1 mM L-glutamine. Synchronization was achieved by serum starvation for 72 h; synchronous growth was induced by re-addition of fully supplemented medium.

## Site directed mutagenesis and recombinant protein expression in NIH3T3 cells

For site directed mutagenesis at K1079 (Human K1083) a PCR based strategy was used. Primers with required mutation (R/Q) were designed and pFastBac™1 vector carrying Rbl2 gene was used as a template for the PCR reaction. After the PCR reaction parental DNA strand (wild type; WT) was digested directly by adding 1ul of Dpn1 enzyme to the PCR amplicon. Dpn1 specifically digest methylated or semi-methylated DNA. Dpn1 digested PCR product was used to transform E. coli strains. Mutants were confirmed through direct sequencing of DNA from bacterial cultures.

For stable expression of mutant and wild-type FLAG-tagged Rbl2 proteins in NIH3T3 cells, an inducible expression system *Rheoswitch*™ system from NEB was used.

## Cell cycle analysis

To analyse cell cycle kinetics of NIH3T3 cells stably expressing mutant Rbl2 proteins, fluorescence activated cell sorting (FACS) analysis was performed as previously described [16].

NIH3T3 cells were arrested in $G_0$ by serum starvation for 72 h, after which cells were re-stimulated by adding fully supplemented media. Cells were harvested later at every 2h intervals and were washed with PBS and fixed with 70% ice-cold ethanol for at least 1h. Cells were collected by centrifugation and washed with PBS. After washing cell pellet was collected by centrifugation at 2,000 rpm for 5 min and digested with (10 ug/ul) RNase-A solution for 30 min at room temperature. 400 ul FACS-buffer (0.05 mg/ml propidium iodide in PBS; pH 7.4) were added to the RNase-A digested cell suspension and FACS analysis was performed. The propidium iodide (PI) fluorescence of individual nuclei was measured using a Flow cytometer (Becton and Dickinson, Mountain View, CA, USA). The nuclei traverse the light beam of a 488 nm Argon laser. A 560 nm dichroic mirror and a 600 nm pass filter (bandwidth 35 mm) were used for collecting the red fluorescence due to PI staining. The forward (FSC) and side (SSC) scatter were simultaneously measured. All data were recorded using specific FACScan research software and further processed with FLowjo program.

### *In vitro* phosphorylation assay

Phosphorylation of recombinant Rbl2 proteins (full length, truncated or mutated versions) was performed by mixing 200 ng of recombinant protein (3 ml) with 10 ml of buffer K (30 mM Tris- HCl, pH 8.0; 2 mM MgCl2, 30 mM NaCl, 0.2 mM dithiothreitol, 1 mM EGTA), 50 mM non-radiolabelled ATP, 20 ng CDK4/ CycD1 (Proquinase GmbH, No 10142-0143-1) and 10 mCi of gamma32P-labelled ATP. For pre-phosphorylation with subse- quent acetylation radioactive gamma32P-labelled ATP was omit- ted. The assay mixture was incubated for 30 min at 37uC. The reaction was terminated by chilling on ice and either addition of SDS-sample buffer (for further SDS PAGE and autoradiography) or by washing the beads 3 times with an excess of buffer P for subsequent acetylation reaction.

### Protein file preparation and molecular docking

The structure of Cyc-D1/CDK4 complex was retrieved from protein data bank (entry code: 2W96). The C-terminal domain of wild-type Rbl2 and its mutated models i.e. K1083R, K1083Q, and K1083D were modelled through *ab initio* modelling server "Quark". Predicted models were validated by plotting rampage and analysing their phi, psi ($\varphi$, $\Psi$) angles.

Both Cyc-D1/CDK4 complex (receptor) and C-terminal domain of Rbl2 (ligands; wild-type and mutated variants) were subjected to blind molecular docking using GRAMM-X Protein-Protein Docking Web Server v.1.2.0 [17, 18], and a total of 10 alternative binding models were generated. The global search was performed using fast Fourier transform (FFT) method by keeping both receptor and ligand as rigid body. The best models were taken for further analysis and for molecular dynamics simulation study to understand their dynamical behaviour.

### MD simulations (MDS)

For MD simulations, the ff14SB amber force fields were used for proteins (Cyc-D1, CDK4 and Rbl2, mutated and wild-type). The xleap program [19] was used to create topology/parameter (.prmtop) file and a coordinate (.inpcrd) file of protein complexes. TIP3P water box was used to explicitly solvate the system. A standard protocol was followed to energy minimize and equilibrate the systems prior to large scale sampling from MD simulations using NAMD [20] tool. It was ensured that both temperature and total energy of each system were converged before sampling. The periodic boundary conditions (PBCs) were used throughout 20ns long MD simulations for each system. The SHAKE algorithm [21] was used to control/fix all bonds involving hydrogen; the time step was set to 2fs. The cut-off for *van der* Waals and electrostatics was set to 12 Å with a switching function of 2 Å. Particle Mesh Ewald (PME) method was

applied for long range electrostatics [22]. The isothermal-isobaric (NPT) ensemble was used to keep the system at 310 K temperature and 1 atm pressure using Langevin thermostat and Nose-Hoover Langevin piston method, respectively. The resulting trajectories were written at every 1ps. For analysis the first 4ns sampled trajectories were again discarded to further ensure any biasness due to energy and temperature non-convergence, if there was any, and the remaining 16ns long trajectory from each system was taken for detailed analysis. Molecular visualization and distance plot generation was performed through Visual Molecular Dynamics (VMD) program [23]. Bio3D module of R program was used to calculate root mean square deviation (RMSD), root mean square fluctuation (RMSF), and radius of gyration (Rg). The details of MMPBSA calculations are given in supplementary materials.

## Results

### C-terminus of Rbl2 is heavily mutated in breast cancer patients

Genomic DNA from freshly obtained biopsies (n = 200) along with their adjacent normal control tissues (ANCT) and blood samples were used to perform SSCP analysis. In the present study, four exons encompassing the pocket and C-terminal domain of Rbl2 gene were analysed comprehensively for any novel or reported germline/somatic mutations using SSCP followed by the direct sequence of suspected individuals (Fig 1) i.e. the individuals that had shown mobility shift in SSCP analysis. Among all the four exons, exon 21 and 22 that encompass C-terminus of the protein were found heavily mutated. From 200 biopsy tissues screened for genetic mutations, total 45 mutations were detected out of which 28 mutations were in exon-21 and 17 in exon-22. These mutations were prevalent in tumor tissues compared to their ANCT controls (Table 1).

Among exon-21 mutations, all the mutations were in germline; out of which 18 were missense and 10 were silent mutations. However, among 17 mutations of exon-22, 15 mutations were in germline while 2 somatic mutations were found. This exon contains 14 missenses and 3 silent mutations, in addition to 01 deletion mutation (S2 Table in S1 File). Third highly mutated exon was exon-19 (11 mutations). Exon-19 encompass characteristic pocket structure of Rbl2. This exon carries 8 germline and 3 somatic mutations, out of these 7 mutations were missense, 01 silent and 03 of the mutations were insertion (Fig 2).

Out of 11 mutations present in **Exon-19**, 10 were novel, while 01 reported mutation at g.46521 (T>C) were found to be a T>G alteration (Fig 2). The exon-19 mutations were more frequent in well differentiated (GIII) and moderately well differentiated (GII) tumors. ˜37% (74/200) of breast tissues and ˜35.5% (71/200) of blood samples were found mutated for this exon, which is alarmingly high. Statistically, the difference of mutations was found significant (p <0.0001) among both blood and tumor tissues compared to their ANCT controls with an odds ratio of 15.18 and 22.91 respectively (Table 1 & S3 Table in S1 File). As the integrity of this exon is pivotal for an intact pocket domain and its association with other proteins, mutations in this exon may have serious consequences.

**Exon-21** was the most highly mutated exon in Rbl2 gene. 94 out of 200 samples that account for 47% of the study cohort (blood) were found mutated for this exon, whereas 50% (100/200) tissue samples were also positive for this exon mutations. The mutations in this exon were statistically significant ($p <0.0001$) in breast cancer patients compared to their ANCT controls with odds ratio of 15.24 and 14.39 for blood and tissues respectively (Table 1), which indicates a significant degree of prognosis attributed with the normal working of this exon. Details of these 28 germline mutations are given in S4 Table in S1 File. This exon ends at codon 1083-AAG that codes for a lysine residue, which is part of one of the two nuclear localization signals (NLS) [1081]PSKRLR[1086] in this protein. Recently, this residue has been reported as a candidate for

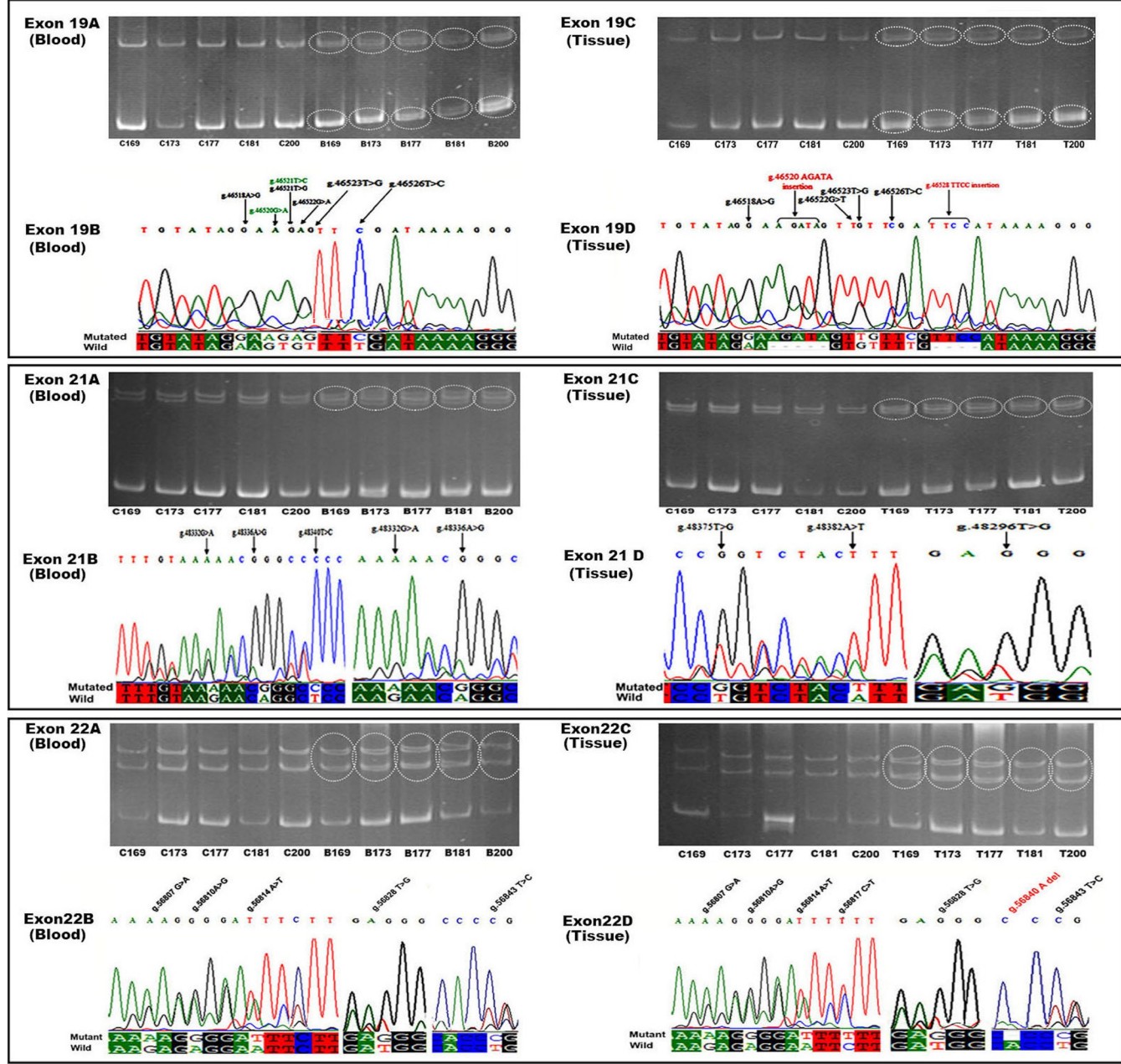

**Fig 1. Mutation profiling of Exon 19–22 of *Rbl2/p130* gene among breast cancer patients.** Each section represents individual exon from 19–22. Panels (A) represent 10% SSCP gel high-lilting mobility shift (circles) in PCR amplicons. latter 'C' represents controls, whereas 'T' represents tissue samples. Panel (B) show the chromatogram of sequencing with arrows indicating the position of mutation.

posttranslational acetylation in a cell cycle dependent manner [11, 12]. This codon at position g.48458 A>G was found mutated to an (AGG) arginine in 102 (51%) blood and 118 (59%) tissue samples (Fig 2). This mutation was significantly ($p < 0.0001$) present in breast cancer patients. This suggests that K1083 mutation might have pathogenic implications resulting in tumor development. Moreover, mutation of this residue has been shown to perturbs its acetylation potential and may have serious consequences in its binding with Cyc/CDK complexes. This highlights significance of this mutation in determining overall prognosis of the disease.

**Table 1. Mutation frequencies in various exons of *Rbl2/p130* gene among various breast cancer cohorts.**

| Exon | | Tumor Grades (n = 200) | | | Diseased (n = 200) | Controls (n = 200) | Odds ratio | CI 95% | | *p* value |
|---|---|---|---|---|---|---|---|---|---|---|
| | | GI | GII | GIII | | | | lower | upper | |
| **19** (g.46510-46615) | No. of times exon mutated in blood | 21 | 27 | 23 | 71 | 7 | 15.18 | 6.77 | 34.04 | <0.0001 |
| | No. of times exon mutated in tissues | 22 | 29 | 23 | 74 | 5 | 22.91 | 9.01 | 58.23 | <0.0001 |
| **21** (g.48295-48459) | No. of times exon mutated in blood | 29 | 31 | 34 | 94 | 11 | 15.24 | 7.81 | 29.73 | <0.0001 |
| | No. of times exon mutated in tissues | 32 | 34 | 34 | 100 | 13 | 14.39 | 7.69 | 26.92 | <0.0001 |
| **22** (g.56754-58273) | No. of times exon mutated in blood | 29 | 33 | 36 | 98 | 7 | 26.49 | 11.86 | 59.16 | <0.0001 |

**Exon-22** was also found severely mutated in breast cancer patients. ~50% (100/200) of breast tissue samples were found positive for this mutated exon. The residues I1073 and R1116 entailing previously reported acetylation sites K1072 and K1115 were also found mutated in this exon. The occurrence of this mutated exon was found significant ($p <0.0001$) in breast cancer patients, which again highlights functional significance of intact C-terminus of the protein. Moreover, higher odds ratio: 26.49 (blood) and 9.53 (tissue) associated with mutated exon 22 also suggests a key role of this exon in determining overall prognosis (Table 1 & S5 Table in S1 File).

## K1083 mutation is associated with survival outcomes of breast cancer patients

Among various Rbl2 mutations K1083R showed strongly positive association with tumor grads, whereas it had a mildly positive association to cooccur with S1080I and T1099P (Fig 3A). On the contrary it was found negatively associated with R1086N mutation. This mutation has no significant association with histopathological types of breast cancer as well as with Rbl2

A)

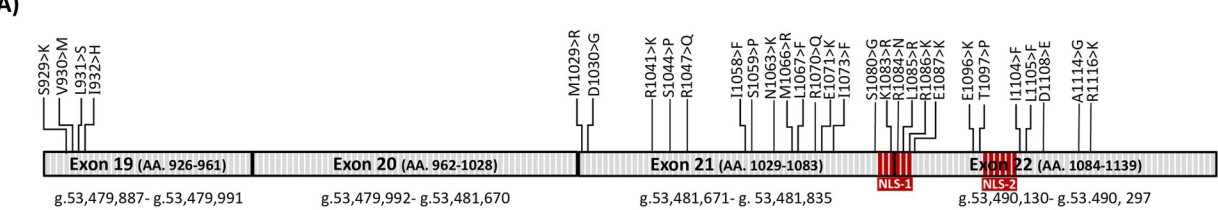

B)

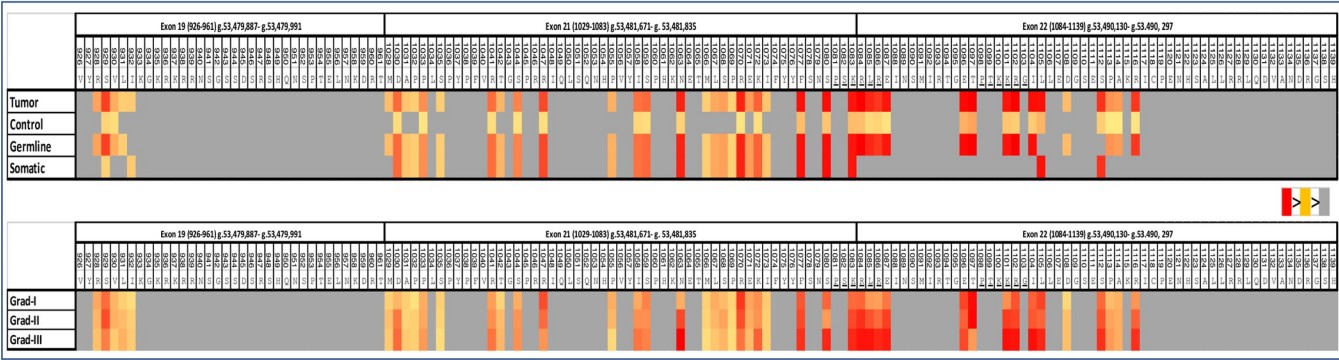

**Fig 2. Association and hazard ratio analysis of various Rbl2 mutation.** A) Correlation plot of various Rbl2 pocket mutations among each other as well as disease grad, histopathological types and Rbl2 transcript expression; B) Forest plot of OR calculated for various Rbl2 pocket domain mutations.

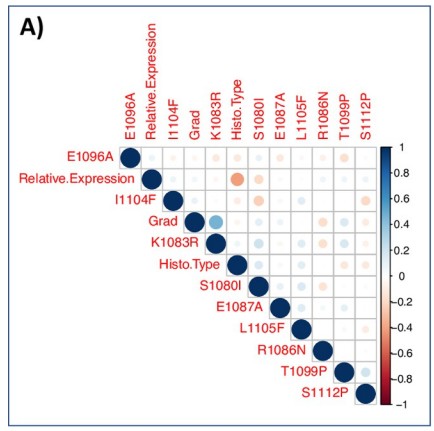

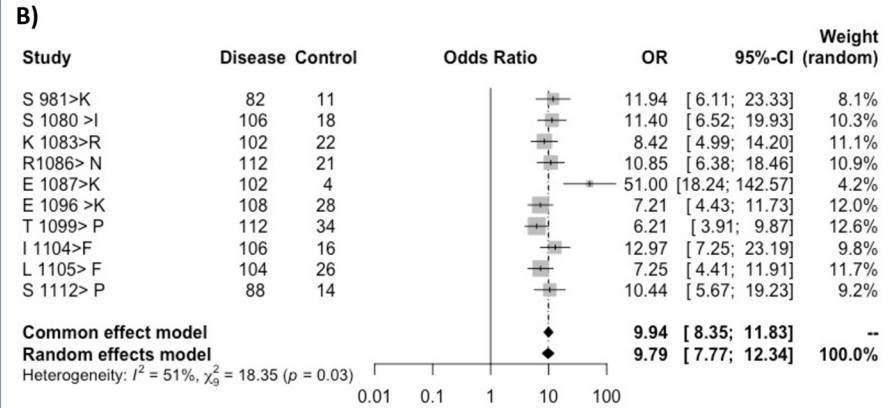

**Fig 3. Survival analysis of Rbl2/p130 C-terminus mutations in breast cancer.** Individual panel represents survival curve calculated through Kaplan Meier analysis for S1080, K1083, R1086, T1099 and I1104 mutations. Among all the five mutations, patients with K1083R mutation exhibited poor survival outcome.

transcript expression. Only the mutation at S1080I had a negative association with Rbl2 transcript expression. Risk analysis of various pocket domain mutations are shown in Fig 3B.

We also analysed survival significance of mutations in or around NLS (S1080, K1083, R1086, T1099 and I1104) through Kaplan Meier curve. Among all the five mutations, patients with K1083R exhibited reduced survival outcome compared to their ANCT controls and this difference was statistically significant (p>0.003; Fig 4). This represents clinical significance of this mutation in overall prognosis of breast cancer patients. However, other mutations (except at R1086) when co-occurring with K1083, could also affect survival outcome of the patients (Fig 5), which is an indication that K1083 may be a driver mutation that confers survival advantage in breast cancer patients.

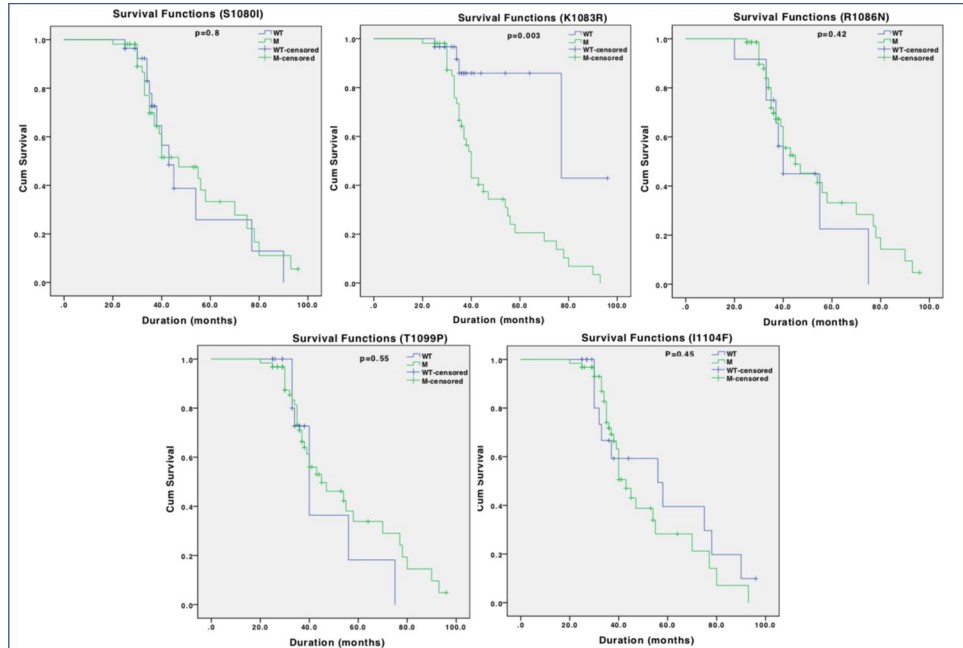

**Fig 4. Survival analysis of Rbl2/p130 C-terminus mutations in breast cancer.** Individual panel represents survival curve calculated through Kaplan Meier analysis for S1080, K1083, R1086, T1099 and I1104 mutations. Among all the five mutations, patients with K1083R mutation exhibited poor survival outcome.

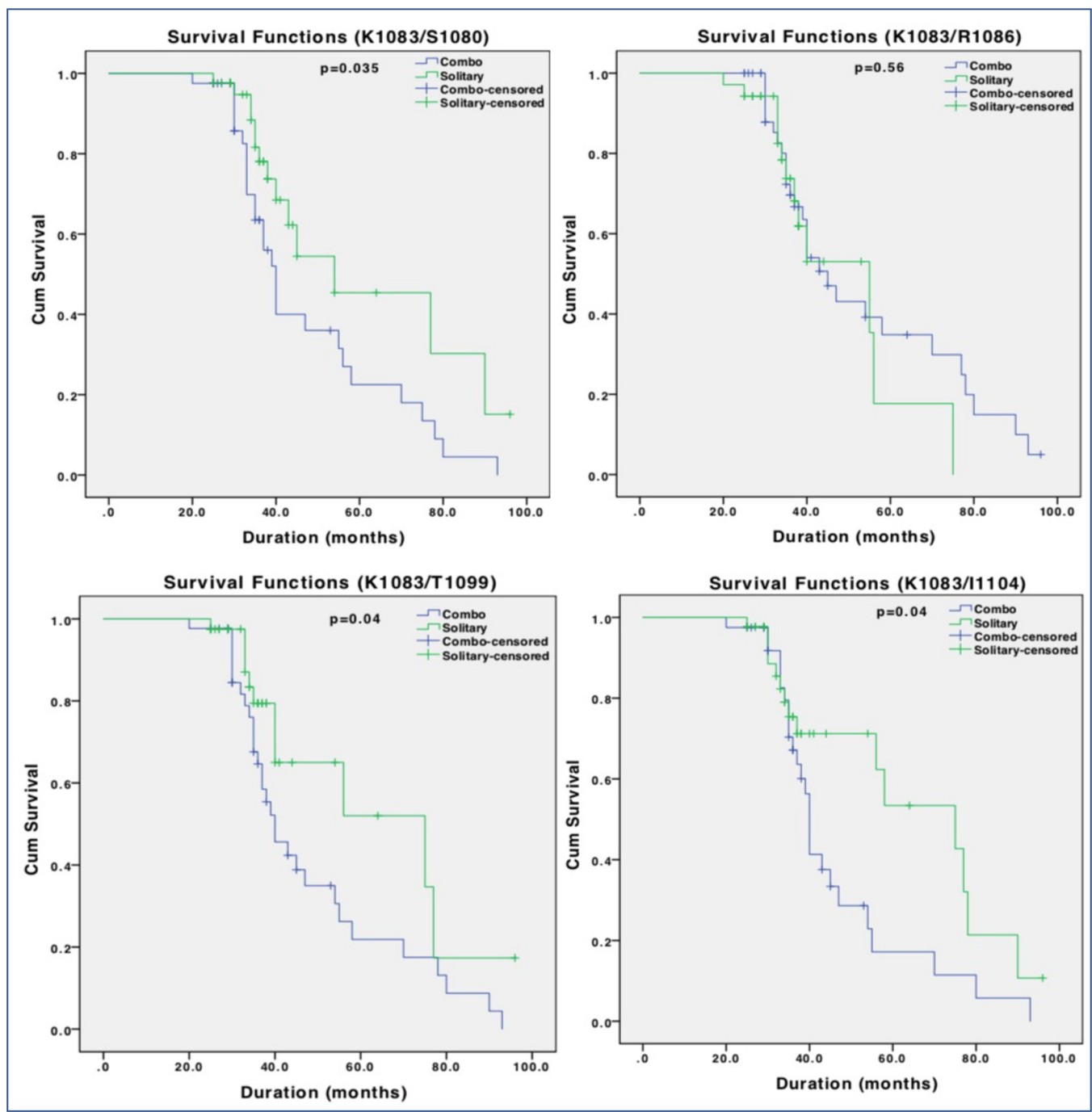

**Fig 5. Impact of Rbl2/p130 C-terminus mutations on survival.** Survival analysis for patients carrying K1083 mutation only (Solitary) and with other mutations (S1080, R1086, T1099 and I1104) was performed through Kaplan Meier. Except R1086, all other mutations when co-occurring with K1083 were associated with poor survival outcome.

## K1083 mutants exhibit perturbed cell cycle events

We investigated the effects of Rbl2 mutations on cell cycle kinetics of NIH-3T3 cells by FACS analysis. Cells expressing wild type as well as mutants (R/Q) at K1079 (human K1083) and mock cells (vector w/o Rbl2 gene) were arrested in $G_0$ and were harvested after every 2h interval.

**Table 2. Cell cycle analysis of NIH3T3 cell lines.** Percentage of cells present in different cell cycle stages is given in h interval after serum restimulation.

| Cell Line | Cell cycle stage (%) | 0'h | 2'h | 4'h | 6'h | 8'h | 10'h | 12'h | 14'h | 16'h | 18'h | 20'h | 22'h | 24'h | 28'h | 32'h |
|---|---|---|---|---|---|---|---|---|---|---|---|---|---|---|---|---|
| NIH3T3 | G1 | 92.44 | 83.60 | 82.76 | 83.72 | 81.10 | 77.33 | 63.11 | 49.06 | 39.21 | 39.44 | 45.04 | 59.96 | 61.99 | | |
| | S | 3.37 | 7.36 | 8.50 | 8.90 | 11.95 | 15.84 | 29.69 | 42.32 | 46.3 | 21.29 | 16.37 | 13.86 | 16.62 | | |
| | G2/M | 4.01 | 8.17 | 7.87 | 6.95 | 7.16 | 6.85 | 6.71 | 8.61 | 14.19 | 38.67 | 38.69 | 26.00 | 21.55 | | |
| Mock NIH3T3 | G1 | 89.19 | | 85.63 | | 78.19 | | 76.32 | | 65.54 | | 73.85 | | 65.17 | | |
| | S | 7.89 | | 11.09 | | 17.03 | | 18.90 | | 26.66 | | 17.69 | | 18.28 | | |
| | G2/M | 2.74 | | 2.81 | | 3.46 | | 4.06 | | 6.54 | | 8.29 | | 16.54 | | |
| p130 (Q) | G1 | 56.52 | | 53.68 | | 54.23 | | 51.39 | | 48.46 | | 46.97 | | 45.02 | 44.63 | 45.27 |
| | S | 6.04 | | 12.05 | | 10.81 | | 11.50 | | 16.20 | | 16.25 | | 13.58 | 18.85 | 18.28 |
| | G2/M | 37.24 | | 33.30 | | 34.06 | | 36.05 | | 33.70 | | 35.72 | | 40.50 | 35.08 | 35.16 |
| p130 (R) | G1 | 92.32 | | 87.94 | | 88.72 | | 86.82 | | 79.88 | | 73.61 | | 74.58 | 77.16 | 75.86 |
| | S | 6.16 | | 6.85 | | 9.45 | | 10.85 | | 17.79 | | 20.11 | | 18.33 | 17.11 | 16.32 |
| | G2/M | 1.51 | | 5.19 | | 1.82 | | 2.30 | | 2.29 | | 6.26 | | 7.08 | 5.72 | 7.80 |

After 72h serum starvation all the cell lines were enriched with $G_1$ populations (~90%) except K1083Q mutant. Normal NIH3T3, mock cells (vector only) and with Rbl2 (WT) had highest 'S' populations after 16h of serum re-stimulation, whereas K1083R mutant was slower through cell cycle progression at this time point (Table 2 & Fig 6). These cells entered S-phase after 20h of serum re-stimulation. These cells were better arrested in $G_0$ and had a prolonged $G_1$-phase in comparison to wild type NIH-3T3 cells. These cells entered S-phase after 20 h of serum re-stimulation. Moreover, they took about 8 h longer than normal to complete their cell

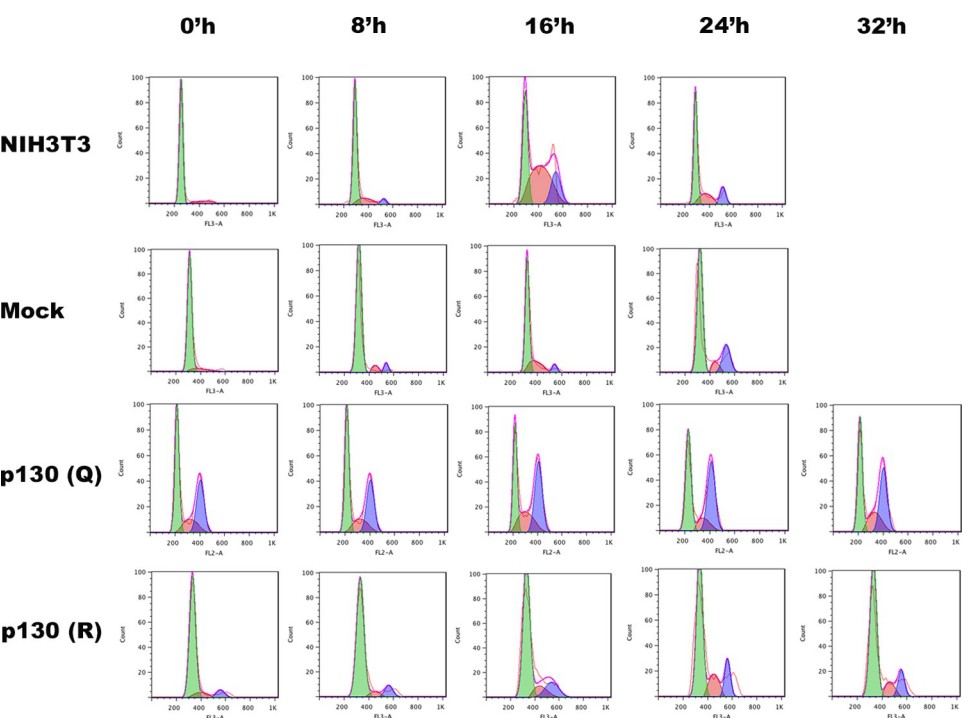

**Fig 6. FACS analysis of NIH3T3 cells stably expressing wildtype Rbl2 and K1083 mutated forms.** Different cell cycle stages are marked with different colors. Cells in G1 are shown in green, S-phase cells are marked with pink, and G2/M phase cells are colored blue.

cycle. K1083Q mutant was not arrested in $G_0$ phase at all and exhibited a non-synchronous pattern at all time points after serum restimulation. At all time-points, almost one-third of the K1083Q mutants' population were present in $G_2/M$ phase. Since these mutants mimic acetylated form of Rbl2, hence might imbalance phosphorylation status of Rbl2 to circumvent $G_1/S$ checkpoint. This highlights a functional relevance of K1083 acetylation and cell cycle control.

## Molecular docking and interaction study

To study kinetics of interaction between residue K1083 and Cyc-D1/CDK4T, the molecular docking and simulation studies were performed. The structure of Cyc-D1/CDK4 complex was retrieved from protein data bank (entry code: 2W96), whereas the C-terminal domain of wild-type Rbl2 and its mutated forms i.e. K1083R, K1083Q, and K1083D were modelled through *ab initio* modelling server "Quark". *In silico* docking analysis of wild-type and three K1083 mutated forms of Rbl2 C-terminus with Cyc-D1/CDK4 complex (S1b and S1c Fig in S1 File) revealed that both wild-type (K1083) and mutant (K1083R) Rbl2 occupied similar region in Cyc-D1/CDK4 complex without having any significant change in interacting residues. However, a detailed interaction study, as displayed in (Fig 7A and 7B), highlighted that these interactions differed in distance and strength due to different geometric structure, pKa values and molecular size of amino acids. The K1083 in the wild-type Rbl2 established a salt bridge with 3 residues (E162, R179, E172) of Cyc-D1. The interaction of K1083 was found interesting with a N-O bond length of 3.85Å which later became very stable during MDS study. On the other hand, K1083R mutant showed strong interactions with two residues (S166, K167) of Cyc-D1 due the presence of guanidinium group of arginine. Simultaneously, it also exhibited strong intra-sub-unit salt bridge interactions with its own E1087 residue, as revealed by MDS. In mutant K1083Q Rbl2 (which mimics acetylated lysine) the important neighbouring residues S1080, P1081, S1082 and R1084 were found exposed to the environment, hence no interaction with Cyc-D1 was noticed. This exposition of important residues to the environment may facilitate phosphorylation of serine or lysine amino acid. The mutant K1083D Rbl2 significantly differed in its interaction pattern, and the protein interacted with CDK4 instead of Cyc-D1. The detail of interactions as obtained from the analysis of molecular docking results is given in Table 3.

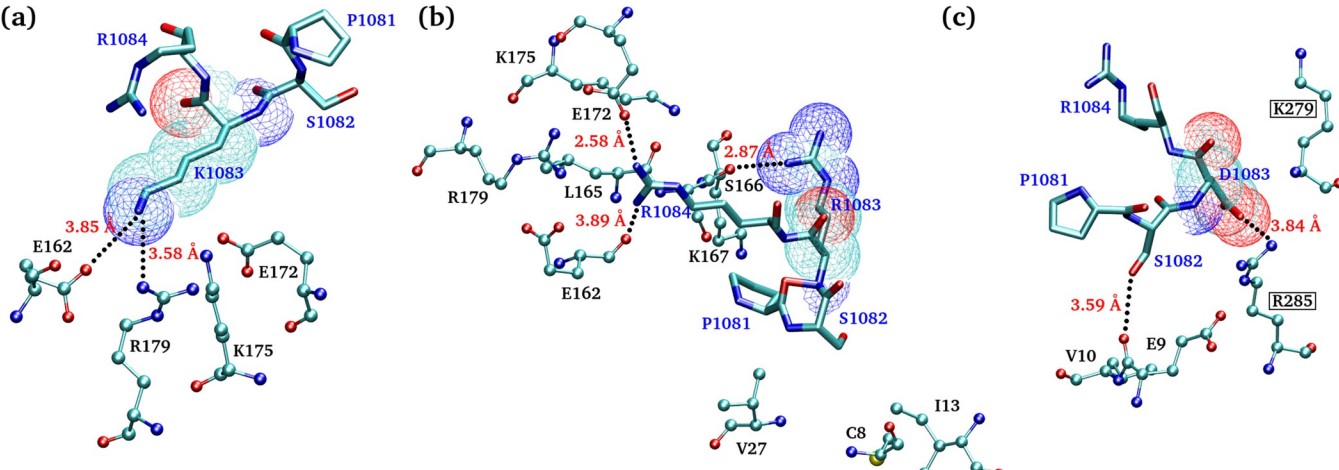

**Fig 7.** Interacting residues of Cyc-D1/CDK4 complex within 5Å of Rbl2 residues P1081, S1082, X1083 and R1084 are shown for (a) Wild-type, (b) K1083R and (c) K1083D ligands. The residues of Cyc-D1/CDK4 are in CPK model and labeled in black while the residues of Rbl2 are drawn as licorice with blue labeled. Hydrogen bonds have been displayed in black dotted lines with corresponding distances in red. Oxygen, nitrogen and carbon atoms are represented by red, blue and cyan colors, respectively.

**Table 3. Some important Interactions between receptor and ligand.**

| Ligand's Residues | Wild Type (K1083) | Mutated (K1083R) | Mutated (K1083Q) | Mutated (K1083D) |
|---|---|---|---|---|
| S68/S1035 | - | - | A−E173/4.52Å; 4.35Å; 4.52Å; 4.74Å | - |
| S1044 | - | - | - | B−T187/4.82Å<br>B-H278/VDW |
| S1068 | B−R285/4.36Å | - | B−R285/3.78Å; 4.82Å | - |
| K1072 | - | - | - | - |
| P1081 | - | A-K167/VDW<br>A-V27/VDW | - | - |
| S1082 | - | A-C8/VDW<br>A-I13/VDW<br>A-V27/VDW | - | B−R285/5Å<br>A−E9/3.59Å; 4.12Å<br>A−V10/4.17Å; 4.23Å/4.76Å |
| X1083 (X = K,R,Q,D) | A−E162/3.85Å<br>A−R179/2.71Å<br>A−E172/3.94Å<br>A−K175/4.95Å | A−S166/2.87Å<br>A−K167/3.24Å; 4.60Å | - | B−K279/4.97Å<br>B−R285/3.84Å |
| R1084 | - | A-S166/VDW<br>A−E162/3.89Å<br>A−K175/3.91Å; 4.13Å<br>A-L165/VDW<br>A−R179/3.03Å; 3.19Å<br>A−E172/2.58Å | - | - |
| T1097 | - | - | - | - |
| S1112 | - | - | - | - |
| K1115 | - | - | - | - |

'A' means cyclin D1 and 'B' means CDK4. 'Text in green background' highlight strong interactions, while in 'grey' moderate interactions, and 'Red background' represents weak interaction.

## RMSD, Rg and RMSF analysis

Further, we investigated the dynamic behavior of Cyc-D1/CDK4–Rbl2 complexes obtained from *in silico* molecular docking experiments, a detailed MDS was performed, and the resultant trajectories were analyzed. Fig 8A & S6 Table in S1 File displays the root mean square deviation (RMSD) of Cyc-D1/CDK4–Rbl2 (wild-type and mutated) protein complexes as observed from 16ns of sampling after discarding the first 4ns of trajectory. Except for K1083R (RMSD$_{Avg.}$ = 2.24Å ± 0.38), all other complexes showed slightly reduced value compared to the wild-type complex (RMSD$_{Avg.}$ = 2.02Å ± 0.39). The K1083Q complex displayed the lowest deviation. However, all four complexes exhibited stable behavior throughout the simulation time indicating overall system stability.

The radius of gyration (Rg) Fig 8B revealed that both Cyc-D1/CDK4–Rbl2 wild-type (black-line) and K1083Q (green-line) complexes had compact molecular structure with Rg$_{avg.}$ = 27.34Å ± 0.13 and 27.10Å ± 0.10, respectively. Although, mutated K1083Q bound with Cyc-D1/CDK4 in different mode, its structural compactness remained intact during simulation.

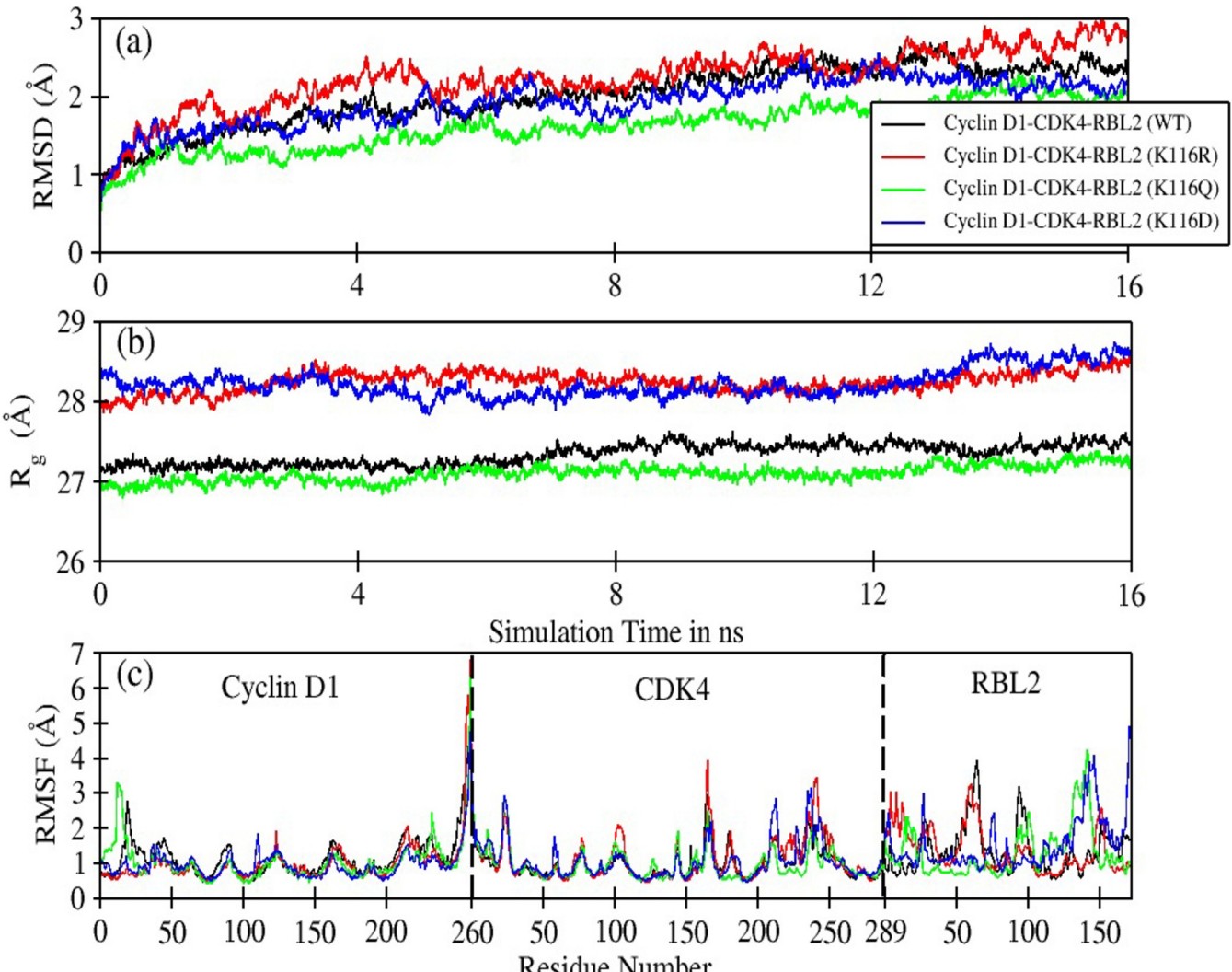

**Fig 8.** Root mean square deviation, RMSD (a), Radius of gyration, $R_g$ (b) and Root mean square fluctuation, RMSF (c) of wild-type (WT) and mutated Cyc-D1/CDK4–Rbl2 protein complexes as observed over the period of 20 ns of sampling. The wild-type Cyc-D1/CDK4–Rbl2 complex is represented as solid black, while mutated (K1083R), (K1083Q) and (K1083D) complexes are represented as red, green and blue colors, respectively. In RMSF graph (c), the amino acid regions indicating Cyc-D1, CDK4 and Rbl2 have been also labelled.

However, K1083R complex, despite having similar basic residue, presented conformational changes that supported intra-subunit salt-bridge formation; thereby may weaken Cyc-D1/CDK4–Rbl2 association. The increase in Rg values by ~1Å reflects that 'R' or 'D' substitution at K1083 produces a slack interface conformation, indicating a slightly instable Cyc-D1/CDK4–Rbl2 complex.

The root means square fluctuations (RMSF) shown in Fig 8(C) suggest that K1083 mutation had a pronounced impact on residues 1–33 (loop-short helix-loop), 58–69 (loop), 90–100 (loop), and 128–148 (loop-sheet-turn-sheet-loop).

The effect on the binding of Cyc-D1/CDK4 complex with mutant R1083 Rbl2 (-128.60 kcal/mol) and K1083Q (-131.97 kcal/mol) was comparable with that of wild-type (-134.26 kcal/mol). K1083D mutation significantly reduced the binding strength with an average binding energy of -109.75 kcal/mol (S7 Table in S1 File).

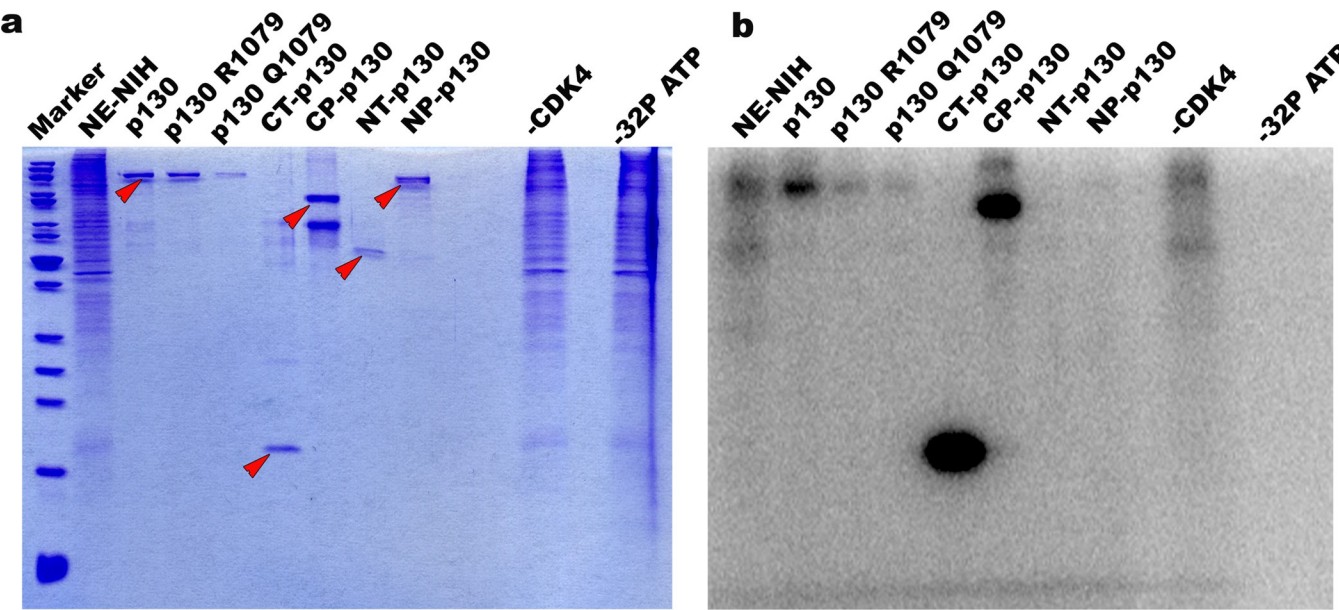

**Fig 9.** *In vitro* **phosphorylation of full length and truncated forms of Rbl2 proteins.** The C- terminus (CT-p130), the C-terminus + pocket domain (CP), the N-terminus (NT-p130) and the N-terminus + pocket domain (NP-p130) were subjected to CDK4 protein kinase assays using $^{32}$P-labelled ATP. Samples were then subjected to SDS-PAGE and subsequent autoradiography. (a) Coomassie stained SDS-PAG. (b) autoradiogram of the Coomassie stained gel. Nuclear extracts of NIH-3T3 (NE-NIH) cells served as a control. Lane marked with -CDK4 contains the nuclear extracts of NIH-3T3 cells and $^{32}$P-labelled ATP without any CDK4, and the lane marked with -32P ATP contain nuclear extracts of NIH-3T3 cells along with CDK4 but without any $^{32}$P-labelled ATP. Red arrows indicate the position of p130 constructs.

### *In vitro* phosphorylation analysis

Full length recombinant Rbl2 and its truncated versions were used to perform *in vitro* kinase assays using recombinant Cyc-D1/CDK4 complexes and $^{32}$P-labelled ATP. Two mutations (R/Q) at K1079 (human K1083) led to lower phosphorylation levels as compared to WT (Fig 9). The truncated N-terminus without C-terminus was unable to be phosphorylated, which is an indication that C-terminus is involved in Cyc-D1/CDK4 interaction, whereas C-terminus alone could undergo this interaction successfully. Moreover, the fact that mutated K1079 had lower levels of phosphorylation also indicate that this residue is pivotal for Cyc-D1/CDK4 association. This corroborate our *in silico* modelling observations that an intact K1083 is indispensable for its proper association to binding partners.

### Discussion

Rbl2 is a nuclear phosphoprotein that has also been reported to undergo acetylation in murine cells [11]. The C-terminal residues K1079, K1068 and K1111 of murine Rbl2 have been identified as stage specific targets of this posttranslational modification. The primary acetylation site K1079 in mouse Rbl2 corresponding to K1083 in human is a located withing highly conserved-PSKRLRE-motif, which is part of the bipartite NLS of the protein. Both acetylated and hyper-phosphorylated Rbl2 forms were shown to localize in the nuclei during $G_1$ to $G_2$ phase, which highlights a mutual interdependence of these modifications in regulating cell cycle. Previously, pRb acetylation was also shown to occur predominantly in the C-terminal part of the protein [14]. The abundance of acetylated lysine/s in the C-terminal domain of pocket proteins indicates some kind of relevance to their functional activities, given the fact that pocket proteins are particularly homologous in the C-terminal domain [24]. Previously, we have demonstrated through *in vitro* experimental data that both of these modifications in Rbl2 protein are mutually inter-dependent [12]. The de-acetylated forms of

Rbl2 were not readily phosphorylated using CDK4, which raises the possibility of acetylation to be a prerequisite for phosphorylation. Even though Rbl2 is acetylated in cell cycle dependent manner, how acetylated Rbl2 is functionally relevant to cell cycle progression is unclear. Fig 3 illustrates a possible involvement of Rb2 acetylation in cell cycle regulation. As have already been discussed diminishing Rbl2 acetylation, also abrogates its phosphorylation potential; suggests an additional regulatory apparatus for cell cycle control. If that is the case, deregulated Rbl2 acetylation profiles in various tumors may also be expected. Our preliminary data supports this hypothesis, where abnormally high Rbl2 acetylation in tumor tissues has been observed [25]. Nonetheless, the putative role of Rbl2 acetylation in carcinogenesis remained elusive.

We report a novel mutation at position g.48458 A>G altering lysine 1083 into an arginine. This transition is particularly important because this K is the target of cell cycle dependent acetylation hence could have pathogenic roles. A transition from lysine (K) to arginine (R) does not change overall positive change of the side chain but may perturb acetylation potential of this motif. Moreover, in the light of earlier observations where this 'K' was mutated to 'R' or 'Q' through site directed mutagenesis and acetylation potential of 'C-terminus' was severely impaired [12], it is plausible to speculate that a codon alteration at this position may have serious implications in the functional activities of this protein and may lead to carcinogenesis. This assumption is also supported by the fact the patients with this codon change presented poor survival outcome (Figs 3 and 4) and enhanced pathogenicity of breast carcinogenesis in current study.

The tendency of Rbl2 proteins to undergo phosphorylation is impaired in proteins mutated at K1083, which raises the possibility that this residue might be involved in steric interaction with CDKs. Our findings of the *in-silico* docking experiments support this steric interaction between K1083 and CDK4/Cyc-D1 complexes and changing this key lysine residue resulted in the abrogation of this complex formation. Consistent with that are the results of *in vitro* phosphorylation assays, where N-terminus without a C-terminus is unable to interact with CDK4/ Cyc-D1 complex and a mutated C-terminus has weakened its association as complex partner. Our analysis showed that K1083Q mutant (mimicking acetylated form of the protein) exposes the neighbouring S1080, P1081, S1082 and R1084, thus enhancing the possibility of Rbl2 phosphorylation. This supports our previous assumption that an acetylated form of Rbl2 can be readily phosphorylated. It is also worth mentioning here that S1080 of human Rbl2 was previously shown to be the target of CDK4 dependent phosphorylation [26]. These findings indicate that acetylation is an additional control of cell cycle progression.

Our cell cycle analysis data of murine cells stably expressing wild type and mutated forms also supports the fact that perturbed acetylation status may affectively upset cell cycle control, especially in case of K1083Q mutated proteins. The non-synchronous behaviour of these mutated forms highlights a regulatory role of acetylation in cell cycle control. Since K1083Q mutants mimic acetylated form (in reference to steric charge) of Rbl2, they might imbalance phosphorylation status of Rbl2 that resulted in the abolishment of $G_1$/S checkpoint. Hence, they remained predominantly in $G_2$/M phase of cell cycle.

In view of these circumstantial evidence, it is persuasive to thoroughly analyse acetylation and phosphorylation status of Rbl2 proteins in breast cancer patients, to fully understand the disease aetiology. This will help deciding therapeutic strategies for administration of breast cancer patients.

## Supporting information

**S1 File. Supplementary information on mutation analysis and *In-silco* models of mutated Rbl2 proteins.**
(PDF)

**S2 File.**
(PDF)

## Acknowledgments

The authors cordially acknowledge and thankful to all the participants and the supporting staff, PGMI; Leady Reading Hospital Peshawar & Poly Clinic Hospital Islamabad. We are also thankful to all the supporting staff of Cancer Genetics & Epigenetics lab, Biosciences (CIIT), Islamabad for their kind cooperation & availability of research platform.

## Author Contributions

**Conceptualization:** Peter Loidl, Muhammad Saeed.

**Data curation:** Farman Ullah.

**Formal analysis:** Nadia Khurshid, Muhammad Saeed.

**Investigation:** Farman Ullah.

**Methodology:** Peter Loidl.

**Project administration:** Muhammad Saeed.

**Resources:** Farman Ullah.

**Software:** Nadia Khurshid, Qaiser Fatimi.

**Validation:** Qaiser Fatimi.

**Visualization:** Qaiser Fatimi.

**Writing – original draft:** Muhammad Saeed.

**Writing – review & editing:** Peter Loidl, Muhammad Saeed.

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
