## [Decision Letter · Decision Letter 0]

3 Feb 2022

PONE-D-22-02667Mutations in the acetylation hotspots of Rbl2 are associated with increased risk of breast cancerPLOS ONE

Dear Dr. Saeed,

Thank you for submitting your manuscript to PLOS ONE. After careful consideration, we feel that it has merit but does not fully meet PLOS ONE’s publication criteria as it currently stands. Therefore, we invite you to submit a revised version of the manuscript that addresses the points raised during the review process.

We look forward to receiving your revised manuscript.

Kind regards,

Giuseppina Caretti

Academic Editor

PLOS ONE

Journal Requirements:

Reviewers' comments:

Reviewer's Responses to Questions

**Comments to the Author**

1. Is the manuscript technically sound, and do the data support the conclusions?

Reviewer #1: Partly

Reviewer #2: Partly

2. Has the statistical analysis been performed appropriately and rigorously? 

Reviewer #1: I Don't Know

Reviewer #2: Yes

3. Have the authors made all data underlying the findings in their manuscript fully available?

Reviewer #1: Yes

Reviewer #2: Yes

4. Is the manuscript presented in an intelligible fashion and written in standard English?

Reviewer #1: Yes

Reviewer #2: Yes

5. Review Comments to the Author

Reviewer #1: Mutations in the acetylation hotspots of Rbl2 are associated with increased risk of breast cancer

Ullah et al

SUMMARY: The RB-like pocket protein RBL2/p130 is a tumor suppressor and important regulator of cell cycle gene expression programs. It is established that, like other RB-family proteins, RBL2 is regulated by phosphorylation. A recent report also demonstrated a role for ubiquitin in controlling p130. In addition the corresponding author has previously shown the RBL2/p130 is acetylated. However, the impact of these post-translational modifications on RBL2 function and their relevance in disease, remain largely unstudied. The authors identify patient mutations in RBL2 and map these do a previously suggested acetylation site. They show a role for these in cell cycle progression, and potentially in Cyclin D-CDK4/6 interactions. While some points of the writing should be clarified, this article adds our understanding of RBL2 function and is worthy of publication.

Points.

1- There are many instances where acronyms are not defined upon first use. For example, what is SSCP- this should defined this when used first in the abstract. The same is true later when they first describe MDS and RMSD experiments.

2- In the very beginning of the results the authors should more clearly state the source of the material used for SSCP analysis. In the methods, it says fresh tissue was collected with adjacent normal and blood. It is not clear from the manuscript where the mutations are arising and if they are in fact more prevalent in diseases. This should be made more clear. Of note, I am not an expert on judging their frequency in tumor vs normal tissues, and these data should be reviewed by someone who is.

3- Results say direct sequencing was done on suspected individuals. What does this mean? That a potential mutation was identified by SSCP, which was then validated by direct sequencing?

4- The authors say that “45 mutations were detected in these exons out of which 28 mutations were in exon-21 and 17 in exon-22.” How many samples were analyzed to identify these 45 mutations?

5- The cell cycle analysis shown in table 2 is very interesting. Could these flow cytometry plots be shown?

6- Are p130(Q) cells arrested in G2? Could the authors elaborate on why this might be the case?

7- The source of the structure of RBL2-cyclin D show in Fig 6 should eb stated in the results section and not only in the methods. As it reads, in the results, it was unclear if that was modeling of a previous structure or the authors own structural data. This should be made more clear.

8- The source of the material used in the final figures should be shown, to demonstrate how pure that protein was, and it should be made clear in the results what the source of that protein is, and more clearly stated how those experiments are done, analyzed and interpreted. These experiments are also outside of my general expertise and should be reviewed by someone more capable of judging those data.

Reviewer #2: Retinoblastoma like protein-2 (Rbl2) is functionally regulated by phosphorylation and acetylation. The authors have previously demonstrated that lysine 1083 (K1079 in human Rbl2) is a potential target for acetylation but its functional role remains elusive. Thus, they investigated alterations in human Rbl2 gene specifically targeting exons 19-22 harbouring acetylatable residues i.e. K1072, K1083 and K1115 through SSCP in breast cancer patients. The K1083 was found altered into arginine (R) in 51% of the cases but K1072 and K1115 remained conserved. The ‘K1083R’ mutation impairs the acetylation potential of this motif that may result in functional inactivation of Rbl2. These patients also showed poor survival outcome that highlights prognostic relevance of this residue. NIH3T3 cells expressing glutamine (K1083Q) mutated Rbl2 could not be arrested in G1 by serum starvation, whereas cells expressing Rbl2 with K1083R showed prolonged G1 arrest in FACS analysis. This suggests that K1083 acetylation is important for G1/S transition. In addition, the authors performed molecular dynamic studies to analyze kinetics of residue K1083 with Cyc-D1/CDK4. Mutations at K1083 impaired this binding exposing residues S1080, P1081, S1082 and R1084 enhancing the possibility of accelerated phosphorylation. S1080 has previously been reported as a promising candidate of cell cycle dependent phosphorylation in Rbl2. This highlights significance of mutations in the pocket domain of Rbl2 gene in breast cancer, and also strengthen the notion that K1083 acetylation is pre-requisite for its phosphorylation.

The paper presents data that have the potential to elucidate mechanisms by which acetylation might regulate RBL2/p130 phosphorylation and action. However, at this stage the paper is very preliminary and data presented purely correlative. This work would benefit from the inclusion of some more mechanistic studies. For example, Co-IP studies to confirm the role that acetylation sites play in regulating the interaction between RBL2/p130 and the cyclinD1-CDK4 complex would confirm the in silico models. In addition, evaluating the expression of RBL2 mutant in relevant acetylation sites and their effect on cell cycle would be another important experiment to include.

6. PLOS authors have the option to publish the peer review history of their article (what does this mean?). If published, this will include your full peer review and any attached files.

Reviewer #1: No

Reviewer #2: No

---

## [Author Response · Author response to Decision Letter 0]

9 Feb 2022

Dear Editor,

The revised manuscript [PONE-D-22-02667] after making necessary changes in the light of comments given by academic editor and reviewer’s is being submitted. The academic editor and worthy reviewers have highlighted valuable points that highlight their in-depth knowledge and understanding of the subject. Both the reviewers have conceded the merit of the manuscript for publication. We are thankful for their kind comments. Following changes have been made to meet the suggestions of the worthy reviewers.

1. “Please ensure that your manuscript meets PLOS ONE's style requirements, including those for file naming. The PLOS ONE style templates can be found at http://www.plosone.org/attachments/PLOSOne_formatting_sample_main_body.pdf and http://www.plosone.org/attachments/PLOSOne_formatting_sample_title_authors_affiliations.pdf”

Response: Manuscript now meets the journal formatting requirements and all the necessary changes has been incorporated. 

2. “PLOS ONE now requires that authors provide the original uncropped and unadjusted images underlying all blot or gel results reported in a submission’s figures or Supporting Information files”

Response: The raw gel figs have been included in Supporting 2.

Rebuttal reviewer #1:

Worthy reviewer-1 has not demanded any additional experiments to decide the merit of this manuscript. We are thankful for his/her kind opinion that manuscript is “worthy of publication”. A point wise rebuttal to all the comments are as follows.

1. “There are many instances where acronyms are not defined upon first use. For example, what is SSCP- this should defined this when used first in the abstract. The same is true later when they first describe MDS and RMSD experiments.”

Response: The point is well taken and suggestion has been incorporated in the revised manuscript.

2. “In the very beginning of the results the authors should more clearly state the source of the material used for SSCP analysis. In the methods, it says fresh tissue was collected with adjacent normal and blood. It is not clear from the manuscript where the mutations are arising and if they are in fact more prevalent in diseases. This should be made more clear. Of note, I am not an expert on judging their frequency in tumor vs normal tissues, and these data should be reviewed by someone who is..”

Response: As suggested by the worthy reviewer it has been explicitly mentioned in the beginning of results section (page 8) that 

“Genomic DNA from freshly obtained biopsies (n =200) along with their adjacent normal control tissues (ANCT) and blood samples were used to perform SSCP analysis.”

Moreover, it has also been clarified on the same page that 

“These mutations were more prevalent in tumor tissues compared to ANCT controls as has been already highlighted in table 1”

3. “Results say direct sequencing was done on suspected individuals. What does this mean? That a potential mutation was identified by SSCP, which was then validated by direct sequencing?” 

Response: This has been clarified in the updated version of the manuscript. The suspected individuals were the patients that shown mobility shift in SSCP analysis. DNA from such cases were sent for sequencing. 

4. “The authors say that “45 mutations were detected in these exons out of which 28 mutations were in exon-21 and 17 in exon-22.” How many samples were analyzed to identify these 45 mutations?”

Response: This has also been clarified in the updated version, now it reads on page 8 that 

“From 200 biopsy tissues screened for genetic mutations, total 45 mutations were detected out of which 28 mutations were in exon-21 and 17 in exon-22.“

5. “The cell cycle analysis shown in table 2 is very interesting. Could these flow cytometry plots be shown?”

Response: The flow cytometric plots has been added as figure 6 in the updated manuscript.

6. “Are p130(Q) cells arrested in G2? Could the authors elaborate on why this might be the case?”

Response: The observation that noticeable population of ‘K1083Q” mutants remained present in G2/M phase is highlighting the significance of acetylation at this particular residue and this is also thrilling for all of us. In the updated version On page 10 and in discussion page 14, we discussed the possible consequences of this mutation that may lead to perturbed cell cycle. On page 14 it reads as 

“Since K1083Q mutants mimic acetylated form (in reference to steric charge) of Rbl2, they might imbalance phosphorylation status of Rbl2 that resulted in the abolishment of G1/S checkpoint. Hence, they remained predominantly in G2/M phase of cell cycle.”

7. “The source of the structure of RBL2-cyclin D show in Fig 6 should eb stated in the results section and not only in the methods. As it reads, in the results, it was unclear if that was modeling of a previous structure or the authors own structural data. This should be made clearer.”

Response: In revised version it is Fig 7In order to clarify the points following para has been on page 11.

“To study kinetics of interaction between residue K1083 and Cyc-D1/CDK4T, the molecular docking and simulation studies were performed. The structure of Cyc-D1/CDK4 complex was retrieved from protein data bank (entry code: 2W96), whereas the C-terminal domain of wild-type Rbl2 and its mutated forms i.e. K1083R, K1083Q, and K1083D were modelled through ab initio modelling server “Quark”.

8. “The source of the material used in the final figures should be shown, to demonstrate how pure that protein was, and it should be made clear in the results what the source of that protein is, and more clearly stated how those experiments are done, analyzed and interpreted. These experiments are also outside of my general expertise and should be reviewed by someone more capable of judging those data.”

Response: The query has been resolved by mentioning the sources of in silico models used for molecular dynamic studies. It is to bring into notice that the said investigations don’t involve wet lab experiments, where highly purified proteins are required to perform interaction studies.

Rebuttal reviewer 2:

Worthy reviewer-2 has also appreciated the data presented in the manuscript and has positively commented on the scope of data to elucidate cell cycle kinetics in further details. We are really thankful for his/her kind words. However, the reviewer has suggested to perform additional experiment to confirm that changes in C-terminal Rbl2 sequences may hinder its association with cyclinD1-CDK4 complex. This will support the in-silico observations that C-terminus and K1083 in particular is involved in this interaction. 

Response: We’re glad to be able to show that without C-terminus an interaction with cyclinD1-CDK4 is not possible. The N-terminus without an intact C-terminus is unable to undergo phosphorylation in the presence of cyclinD1-CDK4 complex. These in vitro kinase assay supports our in-silico observations that C-terminus is involved in this interaction. Moreover, a mutation at K1083 also weakens this association as is evident from the week phosphorylation status of these protein. The results have been included in the updated version of manuscript and are presented in figure 9. 

We are again thankful for valuable feedback of academic editor and both reviewers.

Submitted for onward processing

Sincerely, 

Muhammad Saeed, PhD

Associate professor

---

## [Editor Report · Decision Letter 1]

1 Mar 2022

PONE-D-22-02667R1Mutations in the acetylation hotspots of Rbl2 are associated with increased risk of breast cancerPLOS ONE

Dear Dr. Saeed,

Thank you for submitting your manuscript to PLOS ONE. After careful consideration, we feel that it has merit but does not fully meet PLOS ONE’s publication criteria as it currently stands. Therefore, we invite you to submit a revised version of the manuscript that addresses the points raised during the review process.

ACADEMIC EDITOR:

Thank you for sending a revised manuscript with your responses to the reviewers' comments. I appreciate your additional experiment on Rbl2 mutants phopshorylation, which further support your data.

I am writing to inquiry if you can address this last comment of Reviewer 2, for which I haven't found a response in your letter.

-In addition, evaluating the expression of RBL2 mutant in relevant acetylation sites and their effect on cell cycle would be another important experiment to include.

Please submit your revised manuscript  Apr 15 2022 11:59PM. If you will need more time than this to complete your revisions, please reply to this message or contact the journal office at plosone@plos.org. Please include the following items when submitting your revised manuscript:A rebuttal letter that responds to each point raised by the academic editor. You should upload this letter as a separate file labeled 'Response to Reviewers'.A marked-up copy of your manuscript that highlights changes made to the original version. You should upload this as a separate file labeled 'Revised Manuscript with Track Changes'.An unmarked version of your revised paper without tracked changes. You should upload this as a separate file labeled 'Manuscript'.If applicable, we recommend that you deposit your laboratory protocols in protocols.io to enhance the reproducibility of your results. Protocols.io assigns your protocol its own identifier (DOI) so that it can be cited independently in the future. For instructions see: https://journals.plos.org/plosone/s/submission-guidelines#loc-laboratory-protocols. Additionally, PLOS ONE offers an option for publishing peer-reviewed Lab Protocol articles, which describe protocols hosted on protocols.io. Read more information on sharing protocols at https://plos.org/protocols?utm_medium=editorial-email&utm_source=authorletters&utm_campaign=protocols.

We look forward to receiving your revised manuscript.

Kind regards,

Giuseppina Caretti

Academic Editor

PLOS ONE
---

## [Author Response · Author response to Decision Letter 1]

10 Mar 2022

Dear Editor,

The revised manuscript [PONE-D-22-02667] was submitted to your office after making necessary changes in the light of comments given by academic editor and reviewer’s. Although we have addressed all the points highlighted by worthy reviewers, somehow, we missed to respond this final comment. We sincerely apology for that. 

Comment:

1. “-In addition, evaluating the expression of RBL2 mutant in relevant acetylation sites and their effect on cell cycle would be another important experiment to include.”

Response: We apricate these comments to explore the effect of mutant Rbl2 expression at various acetylation sites on cell cycle. Current study identified 45 mutations in total and out of these 45 mutations 10 missense mutations were observed in/around NLS (Fig 2A). Three acetylatable residues K1072, K1083 and K1115 were specifically scanned for mutations. Only K1083 was found altered into an arginine (R), Whereas K1072 and K1115 remained conserved although a silent mutation at K1072 was observed in few primary tumors (Fig 2B). The Kaplan-Meir analysis presented in Fig 4 & 5 reflects that only mutation at K1083 resulted in poor survival outcome of patients. 

Since we observed acetylatable residue K0183 only as a clinically meaningful factor for survival analysis, we performed in vitro mutagenesis experiments to explore its impact on cell cycle (Fig 6 and Table 2). 

In this context, we believe that worthy reviewer’s comment has already been addressed. Moreover, cell cycle analysis on K1072 and K1115, although is interesting academically, but might not be clinically significant and is beyond the scope of current manuscript 

Further, In silico analysis revealed that K1072 & K1115 don’t interact with Cyclin/CDK4 complex, again highlighting that these residue might not have clinical impact. In this revised version we have included the table (Table 3) of interaction that further support that why cell cycle analysis of K1083 was performed only. 

We are again thankful for valuable feedback of academic editor and both reviewers.

Submitted for onward processing

Sincerely, 

Muhammad Saeed, PhD

Associate professor

---

## [Editor Report · Decision Letter 2]

16 Mar 2022

Mutations in the acetylation hotspots of Rbl2 are associated with increased risk of breast cancer

PONE-D-22-02667R2

Dear Dr. Saeed,

We’re pleased to inform you that your manuscript has been judged scientifically suitable for publication and will be formally accepted for publication once it meets all outstanding technical requirements.

Kind regards,

Giuseppina Caretti

Academic Editor

PLOS ONE
---

## [Editor Report · Acceptance letter]

21 Mar 2022

PONE-D-22-02667R2 

Mutations in the acetylation hotspots of Rbl2 are associated with increased risk of breast cancer 

Dear Dr. Saeed:

I'm pleased to inform you that your manuscript has been deemed suitable for publication in PLOS ONE. Congratulations! Your manuscript is now with our production department. 

Kind regards, 

on behalf of

Dr. Giuseppina Caretti 

Academic Editor

PLOS ONE